# Review of Drug Therapy for Peripheral Facial Nerve Regeneration That Can Be Used in Actual Clinical Practice

**DOI:** 10.3390/biomedicines10071678

**Published:** 2022-07-12

**Authors:** Soo Young Choi, Jung Min Kim, Junyang Jung, Dong Choon Park, Myung Chul Yoo, Sung Soo Kim, Sang Hoon Kim, Seung Geun Yeo

**Affiliations:** 1Department of Otolaryngology Head & Neck Surgery, College of Medicine, Kyung Hee University, Seoul 02447, Korea; soo904@naver.com (S.Y.C.); kpax1727@naver.com (J.M.K.); hoon0700@naver.com (S.H.K.); 2Department of Anatomy and Neurobiology, College of Medicine, Kyung Hee University, Seoul 02447, Korea; jjung@khu.ac.kr; 3Department of Obstetrics and Gynecology, St. Vincent’s Hospital, The Catholic University of Korea, Suwon 14647, Korea; dcpark@catholic.ac.kr; 4Department of Physical Medicine & Rehabilitation, College of Medicine, Kyung Hee University, Seoul 02447, Korea; famousir@naver.com; 5Medical Research Center for Bioreaction to Reactive Oxygen Species and Biomedical Science Institute, Graduate School, College of Medicine, Kyung Hee University, Seoul 02447, Korea; sgskim@khu.ac.kr

**Keywords:** peripheral nerve, facial nerve, regeneration, drug

## Abstract

Although facial nerve palsy is not a life-threatening disease, facial asymmetry affects interpersonal relationships, causes psychological stress, and devastates human life. The treatment and rehabilitation of facial paralysis has many socio-economic costs. Therefore, in cases of facial paralysis, it is necessary to identify the cause and provide the best treatment. However, until now, complete recovery has been difficult regardless of the treatment used in cases of complete paralysis of unknown cause and cutting injury of the facial nerve due to disease or accident. Therefore, this article aims to contribute to the future treatment of facial paralysis by reviewing studies on drugs that aid in nerve regeneration after peripheral nerve damage.

## 1. Introduction

Peripheral facial nerve palsy is caused by dysfunction of the lower motor neurons of the facial nerves that control facial muscles. This condition can be caused by various etiologies, including trauma, neoplasm, autoimmune disease, and viral infection. Among these etiologies, Bell’s palsy is the most common [1,2]. The facial nerve has an extended and convoluted pathway compared with other cranial nerves; thus, it is vulnerable to damage from various causes [3].

Although facial nerve palsy is not life-threatening, facial asymmetry affects interpersonal relationships, causes psychological and neurological stress, and devastates human life. Thus, many efforts to treat and rehabilitate facial paralysis have been made to reduce socio-economic costs. Therefore, it is necessary to identify the cause of facial paralysis and to provide the best treatment. Although recovery after facial nerve injury may vary depending on individual factors, the most important factor is the degree of initial facial paralysis [4]. 

One of the next most important factors is the drug used. However, controversy remains regarding which drug should be administered after facial paralysis occurs. Many studies have reported the use of various drugs for the treatment of peripheral nerve damage. However, many of these drugs have not yet been validated, and some that have been proven effective in animal experiments are difficult to use in humans. 

Moreover, some drugs are complementary and alternative medicines, whose ingredients have not yet been identified and are not readily available in some countries. Additionally, some drugs cannot be easily used in clinical practice due to their side effects. Therefore, among the many drugs reported to be effective in clinical practice, only a few are used in patients with peripheral facial paralysis.

Accordingly, this review summarizes the drugs that can be used in clinical practice as therapies that have been proposed for the treatment of peripheral nerve damage. This review also discusses their mechanisms of action. These drugs are widely used to treat various diseases and can be effective against facial nerve damage. In actual clinical situations, it would be helpful to select a drug treatment that helps facial nerve recovery while minimizing the side effects of facial nerve damage.

## 2. Medications

### 2.1. Steroids

Inflammation occurs as a result of damage of the facial nerve due to paralysis by a virus or damage by direct physical stimulation [5,6,7]. Inflammation and the resulting edema directly or indirectly affect nerve damage and slow nerve recovery. The anti-inflammatory effects of steroids reduce nerve damage, promote nerve recovery, [8,9]. Moreover, steroids inhibit lipid peroxidation, stabilize nerve membranes, and promote axonal regeneration. A recently published Cochrane review also reported the efficacy of steroids in Bell’s palsy, the most frequent facial nerve disease [10]. In addition, steroids have been shown to promote the regeneration of peripheral nerve damage in various animal experiments in previous studies.

Steroids mediate anti-inflammatory and immunomodulatory actions through four molecular mechanisms. The first is a cytosolic GC receptor (cGCR)-mediated genomic mechanism. The most important factor in the anti-inflammatory response to steroids is genomic action. Steroids freely pass through the cell membrane, bind to the cytosolic glucocorticoid receptor in the cytoplasm, move into the nucleus, and inhibit the transcription of inflammatory mediator genes through nuclear factor kappa B (NF-κB) and activator protein 1 (AP-1) to inhibit interleukins (IL) by inhibiting the synthesis of pro-inflammatory cytokines, such as -1, IL-6, and TNF-α, resulting in anti-inflammatory and immunomodulatory effects. 

The second is a cGCR-mediated non-genomic mechanism. Steroids exert their effects on both genes and non-genes. By binding to the steroid receptor in the cytoplasm, anti-inflammatory proteins, such as chaperones and cochaperones, are secreted, an effect mediated by the inhibition of signaling substances or enzymes in the cytoplasm. Third, a membrane-bound GCR (mGCR)-mediated non-genomic mechanism acts on non-genes to bind to cell membrane steroid receptors. 

In an activated immunological state, cell membrane steroid receptors are overexpressed in immune cells and steroid binding induces immune cell lysis. Fourth, in the nonspecific non-genomic mechanism, high concentrations of steroids can show anti-inflammatory and immunomodulatory actions by directly dissolving them in cell membranes, such as plasma and mitochondrial membranes, and changing their properties [11,12,13]. However, the definitive mechanisms underlying the effects of steroids on nerve damage remain unclear.

The different types of steroids have slightly different effects. Steroids that are commonly used clinically include dexamethasone, methylprednisolone, and prednisolone. Steroids are divided into sex hormones, mineralocorticoids, and glucocorticoids according to their mechanisms of action. Mineralocorticoids and glucocorticoids are produced in the adrenal cortex. Aldosterone is the natural adrenocortical hormone, and cortisol is the natural glucocorticoid. In general, the term steroids refers to glucocorticoids but may contain some degree of mineral corticosteroid effects. Several synthetic glucocorticoids have been developed, which differ in their chemical structures, resulting in various potencies and half-lives (Table 1) [14].

Dexamethasone reduces Wallerian degeneration after peripheral nerve injury and affects myelin debris clearance. In addition to its anti-inflammatory response, dexamethasone also activates various signaling substances, thereby, reducing both the damage caused by free radicals that are released after peripheral nerve damage and the formation of fibrotic tissue [15]. Dexamethasone also reduces aquaporin 1 protein levels, thereby, reducing edema and nerve damage [16]. 

Moreover, topical dexamethasone was shown to be helpful for nerve recovery in peripheral nerve damage, possibly because of myelin sheath thickening and increased axon diameter reported in histological examinations following topical dexamethasone treatment [17]. While both topical and systemic dexamethasone were reportedly effective for peripheral nerve damage, topical dexamethasone was more effective, suggesting its efficacy for facial nerve damage during middle ear surgery [18].

Many studies have reported the benefits of methylprednisolone (MP) in recovery after peripheral nerve damage. One study reported that the administration of MP and ozone after sciatic nerve injury helped nerve recovery by reducing the inflammatory response, perineural granulation, and nerve degeneration [19]. A study using guinea pigs reported that facial nerve damage increased the formation of nitric oxide synthase and nitric oxide in the brainstem, which was delayed by the administration of MP to promote peripheral nerve survival [20]. 

Another study reported similar results, showing that MP reduced nerve damage by reducing nitric oxide and malondialdehyde levels after peripheral nerve damage. Lipid peroxidation is a major factor in malondialdehyde production, while nitric oxide is associated with oxidative stress. In addition, MP increases nerve growth factor and vascular endothelial growth factor during nerve recovery after peripheral nerve injury [21]. MP has also demonstrated different effects depending on the type of facial nerve injury. 

For instance, MP is effective in the recovery of facial nerve damage caused by compression but has no effect on facial nerve palsy and amputation damage caused by herpes simplex virus (HSV) type 1. In one study, MP decreased axonal and myelin degeneration in the compression injury group, while it decreased edema in the HSV type 1 facial nerve palsy group. However, MP was not associated with perineural fibrosis, increased collagen fibers, or Schwann cell proliferation [22]. Similar to dexamethasone, other studies have reported that the topical application of MP is more effective than its systemic application in the recovery of peripheral nerve damage. 

Histologically, topical MP reduces the formation of fibrotic tissue and thickens myelin and axons [23,24]. The local administration of MPs is reportedly more effective in the recovery of peripheral nerve damage to locally slow-release MP compared to the administration of high-dose MP. Topical MP thickened the myelin sheath in mice with damaged sciatic nerves and helped restore peripheral nerve function by increasing the number of nerve fibers and producing a large amount of collagen [25]. 

Two definitive studies have demonstrated the efficacy of this treatment in human Bell’s palsy. In these two prospective studies, prednisolone significantly increased the recovery rate of facial paralysis in patients with Bell’s palsy [26,27] (Table 2).

### 2.2. Statins

Statins lower the blood cholesterol levels by increasing low-density lipoprotein (LDL) receptor expression through the inhibition of 3-hydroxy-3-methylglutaryl coenzyme A (HMG-CoA reductase) [31]. In addition to the primary action of lowering the lipid levels, other effects of statins have also been reported. Conserving endothelial nitric oxide synthase (eNOS) in endothelial cells helped induce the growth of vascular endothelial cells by dilating blood vessels [32]. Moreover, the inhibition of cyclooxygenase-2 inhibits myocyte infiltration and reduces the secretion of metalloproteinases, thereby, reducing plaque vulnerability in blood vessels [33]. 

In animal experiments, statins promote bone formation, which explains the relationship of statins with increasing bone morphogenetic protein-2 (BMP-2) levels, which helps bone formation [34]. The anti-inflammatory effect is one of the main effects of statins and has been reported in several studies. In particular, statins affect inflammation control independently of lowering LDL cholesterol levels [35,36]. Many studies have evaluated how the effects of these statins can be applied in clinical practice, and research results have demonstrated the usefulness of statins even after peripheral nerve damage. Thus, statins may be helpful even after facial nerve injury.

Simvastatin is effective in restoring the function of damaged sciatic nerves in rats. Similar to steroids, rats administered statins showed reduced edema around the nerve damage, myelin debris, and sheets as well as significantly increased blood leukocyte levels [37]. Another study reported that the topical administration of simvastatin using a hydrogel helped restore function in rats with sciatic nerve injury. In this study, simvastatin thickened the nerves during regeneration and increased the expression of several neurotrophic factors (pleiotrophin, hepatocyte growth factor, vascular endothelial growth factor, and glial cell line-derived neurotrophic factor) [38].

The efficacy of recovery from peripheral nerve damage was also reported in a study using atorvastatin. In this study, the systemic administration of atorvastatin in mice reduced damage-associated alterations, including structural disruption, oxidative stress, inflammation, and apoptosis, through the control of several factors [39]. A similar study in rats reported that atorvastatin induced the recovery of peripheral nerve damage and, accordingly, aided the recovery of related muscles [40].

Although these studies revealed that the main effect of statins, lipid reduction, was helpful in peripheral nerve recovery through various other mechanisms, an interesting study showed that simply reducing lipids could help peripheral nerve recovery in rats, wherein lipid depletion promoted peripheral nerve recovery by increasing axonal growth and regeneration [41] (Table 3).

### 2.3. Hormones

#### 2.3.1. Melatonin

Melatonin is a hormone secreted mainly by the pineal gland at night, and its main role is the regulation of the circadian cycle. While it is secreted in small amounts from the retina, gut, skin, platelets, and bone marrow, its systemic effects are insignificant. Many studies have demonstrated various roles for melatonin in the regulation of the circadian cycle. Melatonin is involved in the regulation of blood pressure, the immune response, hemostasis, cell regulation, respiratory chain, and antioxidant defense [42]. A receptor for melatonin was reported in the peripheral vasculature through which melatonin causes vasodilation [43]. 

The role of melatonin in inhibiting inflammation has been studied in association with various diseases [44,45,46,47]. Among these anti-inflammatory effects, the scavenging of NO and free radicals in lymphoid cells by melatonin has attracted attention [48,49]. Melatonin also acts as an antioxidant in the body and has a protective function in cells [50,51,52]. Melatonin also affects the immune system. Recent studies have assessed the possible anti-cancer effects of melatonin [53,54]. These various clinical effects of melatonin suggest the possibility of cell regeneration, proliferation, and protective effects, which are emerging as effective treatments for facial nerve damage.

Animal studies have revealed that melatonin aids in recovery after facial nerve injury. Histological analysis in one study revealed that melatonin decreased axon degeneration, increased collagen fibers, and decreased myelin debris [55]. Another study reported that melatonin reduced lipid peroxidation, axonal injury, and myelin breakdown and increased superoxide dismutase, catalase, and glutathione peroxidase activities in rats with damaged sciatic nerves, thus, promoting nerve recovery. 

In addition, it is reportedly more effective than crush injury in cut injuries with large nerve damage [56]. Another study demonstrated the efficacy of melatonin in an experiment in which melatonin was administered after injuring nerve blood vessels by stripping the epineurium of the sciatic nerve in rats. The findings suggested that melatonin could be used for various damage mechanisms in the peripheral nerves [57]. 

Since melatonin is physiologically secreted in the dark, another study showed that the administration of melatonin in the dark was more effective in recovering damaged peripheral nerves [58]. A similar study using rat nerves reported that melatonin was an important factor in neutrophil activity and reduced the production of myeloperoxidase, which promoted lipid peroxidation and the production of malondialdehyde, a product of lipid peroxidation, and reflects the degree of cellular damage [59].

Another study reported that high-dose melatonin (100 mg/kg) is effective in the functional recovery of peripheral nerve damage. Similar to previous studies, melatonin played a role in preserving the myelin sheath and preventing axonal loss. However, there is no consensus regarding the appropriate melatonin dose for peripheral nerve injury [60]. One molecular study reported that the regeneration mechanism of melatonin is mediated by MT1 and is partly due to the maintenance of sustained ERK1/2 pathway activity [61]. 

Another molecular-level study observed that melatonin increased the expression of not only the melatonin receptor but also those of GAP43 and beta3-tubulin (markers of neurite sprouting in regenerating neurons), and inhibited the activity of calmodulin-dependent protein kinase II. Moreover, the decrease in beta3-tubulin and melatonin receptors was confirmed using luzindole, a melatonin receptor antagonist. 

These results suggest that melatonin promotes nerve regeneration through a receptor-dependent pathway [62]. In one study, the topical administration of melatonin using a 3D melatonin nerve scaffold improved the immune environment by reducing oxidative stress, inflammation, and mitochondrial dysfunction, similar to systemic administration, thereby, aiding in the recovery of peripheral nerve damage [63] (Table 4).

#### 2.3.2. Growth Hormones

Growth hormones (GHs) are secreted by the pituitary gland and are mainly involved in growth by direct action on the body or through insulin-like growth factor-1 (IGF-1). Both hormones reduce the fat mass in the body, promote protein synthesis and bone growth, and are involved in insulin metabolism, thus, regulating the glucose levels. These hormones also improve cardiovascular performance and increase the oxygenation in peripheral organs [64]. IGF-1 passes through the blood-brain barrier (BBB), promotes neurogenesis and synaptogenesis, and protects the cells and nerves from hypoxic damage and chemical toxicity. 

Although the exact mechanisms of GHs and IGF-1 are not known, they affect the cognitive function of the brain by improving the vascular density and glucose utilization [64,65]. These two hormones interact to induce cell hypertrophy, which prevents apoptosis and enables cell division [66]. GHs have stimulatory effects on collagen and bone, which may be effective in treating osteoporosis in postmenopausal women [67,68,69,70]. Due to these characteristics, GH and IGF-1 are being studied and applied in various body regenerative treatments, and are expected to contribute to the progress in the regeneration of peripheral nerves.

A previous study reported increased nerve regeneration after the systemic administration of GH following injury to the median nerve in rats. Histologically, the axon density, axon diameter, and myelin thickness increased, the compound action potential of the innervated muscle increased, and atrophy decreased, resulting in increased muscle function recovery [71]. In a similar study, the systemic administration of GH also promoted axonal regeneration and reduced the muscle atrophy in rats with sciatic nerve injury. GHs also maintained Schwann cell proliferation during prolonged denervation [72]. 

The histological examination of rats with damaged ulnar nerves also showed that GHs histologically increased the myelin levels and decreased fibrosis and granulation [73]. In one study, after sciatic nerve injury in mice, IGF-1 was introduced into the muscle by hydrodynamic injection of IGF-1-expressing plasmid DNA using a biocompatible nonviral gene carrier, a polyplex nanomicelle. Early recovery of sensation in the area distal to the injury was induced via the introduction of IGF-1-expressing pDNA [74]. In another study, locally delivered IGF-1 was effective in nerve regeneration and neuromuscular recovery. In both young and aged animals, IGF-1 significantly improved the axon number, diameter, and density [75] (Table 5).

### 2.4. Carnitine

L-carnitine (levocarnitine; 3-hydroxy-4-N-trimethylaminobutyrate) is synthesized in the living body and is involved in beta-oxidation by transporting fatty acids to the mitochondria [76]. L-carnitine is not taken up by the muscle and heart cells and causes myopathy and heart disease [77]. Carnitine plays a decisive role in maintaining the acetyl-CoA ratio in cells, which helps maintain homeostasis during exercise, ischemia, fasting, and acute stress [78]. Dietary L-carnitine has an anti-aging effect through reducing the generation rate of free radicals, which are the main culprits of aging in the central nervous system. 

In addition, carnitine-fed rats showed improved brain function [79]. Carnitine also affects cognitive function in the human brain. Carnitine passes through the BBB, acts as an antioxidant in the brain, energizes the brain, and helps improve memory and visual-motor coordination [80,81,82]. The antioxidant effect of L-carnitine is also effective in treating male infertility [83]. 

Some studies predicted that L-carnitine could increase bone mass by proliferating osteoblastic cells and promoting collagen synthesis; thus, L-carnitine could be used in the treatment of bone fractures and osteoporosis [84,85,86,87]. Moreover, one study showed that L-carnitine can be helpful in the treatment of pulmonary tuberculosis, while playing a role in the immune response [88]. These key roles in the body, antioxidant effects, and immune modulator roles suggest the positive effects of L-carnitine, even in peripheral nerve damage.

In a rat experiment using carnitine, acetyl-L-carnitine helped nerve regeneration by thickening the myelin sheath in damaged peripheral nerves [89]. In another rat experiment, acetyl-L-carnitine administration after sciatic nerve injury reduced neuronal death and helped peripheral nerve recovery, and high-dose (50 mg/kg/day) treatment was more effective than low-dose (10 mg/kg/day) treatment [90]. Another study that administered acetyl-L-carnitine after damage to peripheral nerves also showed that a dose of 50 mg/kg/day helped in the recovery of peripheral nerves without side effects. In this study, the number of myelinated axons was significantly higher in the acetyl-L-carnitine-treated group, and the myelin and axon thicknesses were greater than those in the untreated group [91]. 

Another study reported that the topical administration of acetyl-L-carnitine was also effective for motor and sensory recovery after peripheral nerve injury [92]. A molecular study reported that acetyl-L-carnitine aided recovery after peripheral nerve injury by preventing the induction of apoptosis, which impaired caspase 3 protease activity and reduced pyknotic nuclei due to the upregulation of the X-linked inhibitor apoptosis protein [93] (Table 6).

### 2.5. Vitamin B12 (Cobalamin)

Vitamin B12 is a part of the vitamin B complex. It cannot be synthesized in the body; thus, it must be ingested through food (mainly animal proteins) [94,95]. Vitamin B12 plays a key role in fat, protein, and monohydrate metabolism, and is essential for cellular respiration [96]. Various diseases occur when the vitamin B12 intake is insufficient. Insufficient vitamin B12 intake may lead to megaloblastic anemia, pancytopenia, or hyperhomocysteinemia, which are closely associated with cardiovascular diseases [97,98]. 

When vitamin B12 levels are insufficient, demyelination and degeneration occur in the nervous system, leading to conditions such as optic atrophy, anosmia, glossitis, paresthesia, and cognitive defects [99]. Vitamin B12 regulates growth factors, macrophage function, and the coagulation system, thus, highlighting the potential of vitamin B12 in alleviating severe inflammatory conditions [100]. Vitamin B12 is also an antioxidant [101].

A study of damaged tibial nerves in rats reported that vitamin B12 was effective for nerve recovery, suggesting that vitamin B12 helps restore peripheral nerve function by reducing Wallerian degeneration [102]. Another study measured vitamin B12 levels using enzyme-linked immunosorbent assays following damage to the sciatic nerve in mice, suggesting the potential usefulness of this treatment [103]. A study in humans with peripheral neuropathy showed that intravenous administration of high-dose vitamin B12 was effective in peripheral neuropathy and chronic axonal degeneration without side effects [104]. 

A study using rats showed that systemic administration of mecobalamin helped recovery in rats with sciatic nerve damage. In the mecobalamin-treated group, histological examination revealed myelin sheath thickening and decreased innervated muscle atrophy. In addition, real-time polymerase chain reaction analysis showed that mecobalamin increased the mRNA expression of growth-associated protein 43 in nerve tissues and increased the mRNA expression of neurotrophic factors (nerve growth factor, brain-derived nerve growth factor, and ciliary neurotrophic factor) [105]. A study on the involvement of vitamin B12 in nerve regeneration at the molecular level reported that vitamin B12 helps peripheral nerve recovery by increasing Erk1/2 and taste activity through the methylation cycle [106] (Table 7).

### 2.6. Ginkgo Biloba

Ginkgo biloba is commonly used as a therapeutic agent for early stage Alzheimer’s disease, vascular dementia, peripheral claudication, and tinnitus of vascular origin [107,108,109]. Ginkgo is a neuroprotective agent, antioxidant, free-radical scavenger, membrane stabilizer, and inhibitor of platelet-activating factor [110,111,112,113]. In in vitro studies, ginkgo prevented neuronal death induced by hypoxia, nitric oxide, and cyanide [114,115,116]. Moreover, ginkgo plays a role in scavenging free radicals. It not only directly removes free radicals but also helps in the upregulation of antioxidant enzymes and proteins [117,118]. Studies on its efficacy against peripheral nerve damage are also being actively conducted.

In rats with sciatic nerve injury, the systemic administration of ginkgo thickened the nerve diameter and increased the number of myelinated fibers. Mice treated with ginkgo showed increased expression of CD34, a marker of axon angiogenesis, and angiogenesis-related genes (Vegf, SOX18, Prom 1, and IL-6) [119]. In a study on how the recovery of peripheral nerve damage proceeds depending on the dose of ginkgo, the administration of ginkgo at a high dose (200 mg/kg/day) was more effective for nerve regeneration than administration at a moderate (100 mg/kg/day) or low dose (50 mg/kg/day) [120]. 

The topical application of ginkgo is also effective in repairing peripheral nerve damage. A study on damaged sciatic nerves in rats demonstrated better recovery after the topical administration of ginkgo. Similar to other studies, the ginkgo-administered group showed a significantly increased number of myelin axons and significant functional recovery with electromyography [121] (Table 8).

### 2.7. Coenzyme Q10

Coenzyme Q10 (CoQ10) is present in the mitochondria of all cells and generates energy in the form of ATP using oxygen. CoQ10 also has antioxidant properties and prevents the cell destruction caused by the free radicals formed during excessive exercise or energy generation. The main functions of CoQ10 are as follows. First, it is distributed in the inner mitochondrial membrane and promotes electron transport chain processes to generate cellular ATP. Second, it maintains moisture in the cell membrane and reduces vitamins E and C so that they can be recycled as antioxidants. Third, it is a powerful antioxidant that dissolves lipids and prevents cell damage by inhibiting lipid peroxidation of the cell membranes. 

Therefore, CoQ10 is closely related to the mechanisms underlying aging-related diseases and physiological aging, with many related research results. Mitochondria not only generate free radicals in the process of obtaining energy but are also the organelles most damaged by free radicals. CoQ10 is an important component of energy generation and uncoupling proteins in the mitochondrial cell membrane and is an essential element for maintaining mitochondrial function. Therefore, a lack of CoQ10 leads to decreased mitochondrial function and accelerated aging [123,124].

Studies have investigated the role of CoQ10 in peripheral nerve regeneration. One study randomly divided Sprague–Dawley albino rats into two groups to investigate the effect of CoQ10 on regeneration in facial palsy after facial nerve injury. The experimental group was intraperitoneally administered CoQ10 (10 mg/kg) for 30 days and then compared to the control group administered saline solution (1 mL/day). Compared with the control group, the CoQ10 group showed greater neurological improvement (*p* = 0.05). 

Moreover, light microscopy revealed significant differences in the vascular congestion, macrovacuolization, and myelin thickness between the two groups (*p* < 0.05) [125]. Crush damage to the sciatic nerve showed results similar to those of the facial nerve. After compression injury in 45 male Wistar rats weighing 160–180 g, the intraperitoneal administration of CoQ10 (10 mg/kg/day) was compared with the non-administered group. The number and diameter of myelinated fibers were significantly increased (*p* < 0.05). Thus, intraperitoneal administration of CoQ10 after compression injury improved sciatic nerve regeneration [126] (Table 9).

### 2.8. Nimodipine

Nimodipine is a calcium channel blocker belonging to the dihydropyridine class of drugs. This drug protects cells by inhibiting the increase in intracellular calcium, which causes nerve damage by blocking L-type calcium channels. Additionally, nimodipine may promote nervous system regeneration by improving the circulation in damaged nerves and axons. The mechanism by which nimodipine restores nervous system damage has not yet been fully elucidated. 

Nimodipine plays a role in increasing nerve regeneration after sciatic nerve injury and in preventing aging-associated degeneration of the nervous system in rats. Blocking L-type calcium channels increases intracellular calcium levels, which causes nerve damage. Thus, nimodipine may promote nervous system regeneration by protecting cells through the inhibition or improvement of the circulation of damaged nerves and axons [127,128].

In one study on Wistar rats, the control group received a placebo, and the experimental group received a food tablet containing 1000 ppm of nimodipine after facial nerve dissection and anastomosis. The number of sprouted motoneurons in the nimodipine group was twice that in the control group [129]. Another animal study showed the beneficial effects of nimodipine on the preservation and restoration of facial and auditory nerve function after vestibular schwannoma surgery. Nimodipine treatment has also been attempted in patients with peripheral facial nerve palsy following maxillofacial surgery. 

After maxillofacial surgery, nimodipine was orally administered to 13 patients who developed moderate-to-severe peripheral facial nerve paresis. All patients showed restoration of facial nerve function within two months [130]. Electromyography (EMG) performed in adult Sprague–Dawley rats that were administered nimodipine (6 mg/kg/day) after crush injury of the facial nerve showed recovery of electrical conductivity in the nimodipine group 20 days after injury. Histological findings of the facial nerve also showed clear recovery of myelination and decreased numbers of infiltrating cells with a reduced inflammatory response [131] (Table 10).

### 2.9. Ozone

The known clinical indications for ozone are as follows: (1) Arteriovascular disease: treatment activates red blood cell metabolism and releases oxygen. (2) Skin ulcers: treatment disinfects wounds, cleans wounds, and promotes wound healing. (3) Diseases of the colon: treatment of colitis, disinfection of fistula wounds, immunity enhancement, and anti-inflammatory action. (4) Infection by viral diseases: treatment improves immunity. (5) Adjuvant treatment of cancer patients: treatment improves immunity. (6) Geriatric diseases: treatment results in antioxidant action and immune system activation. (7) Rheumatoid disease (osteoarthritis): treatment provides an anti-inflammatory effect and activates the antioxidant and immune systems. (8) Dental field: treatment provides disinfection, wound cleaning, and wound healing [132].

Several animal studies have investigated the usefulness of ozone therapy in peripheral nerves. One study randomly divided 14 Wistar albino rats into a control group that received saline treatment and an experimental group that received ozone treatment after crush injury to the facial nerves. The ozone-treated group had a significantly lower stimulus threshold than that of the control group. Significant differences were also observed between the ozone and control groups in terms of vascular congestion, macrovacuolization, myelin thickness, axonal degeneration, and the myelin microstructure. 

Therefore, ozone therapy demonstrated beneficial effects on the regeneration of crushed facial nerves in rats [133]. A study evaluating the efficacy of ozone treatment after a cutting injury to the sciatic nerve in 100 Wistar albino rats reported more myelinated nerve fibers in the ozone-treated group. In addition, plasma superoxide dismutase, catalase, and glutathione peroxidase activities differed significantly in the ozone-treated group along with a functional improvement. The results of this study suggested ozone therapy as a promising alternative for improving peripheral nerve damage [134] (Table 11).

### 2.10. Antiviral Agents

Antiviral agents, such as acyclovir and valacyclovir, are representative drugs used to treat HSV and varicella zoster virus (VZV). Acyclovir is an acyclic nucleoside and nucleotide analog that interferes with the elongation of the viral genome during replication, which is conducted by viral DNA polymerase. Valacyclovir and famciclovir are nucleic acid analogs similar to acyclovir with a covalent mechanism of action that interferes with the function of viral DNA polymerase. Each antiviral agent differs primarily in terms of the bioavailability, half-life in the body, and dosing [135,136].

If the peripheral nerves, including the facial nerve, are infected with a virus, they may be damaged, and symptoms, such as paralysis may manifest. Bell’s palsy and Ramsay–Hunt syndrome are typical diseases that cause paralysis due to viral damage to the facial nerve. Various animal experiments have revealed that antiviral drugs are effective against infections, such as by HSV, that cause these diseases; however, their effectiveness in human Bell’s palsy remains controversial [26,27,137]. Conversely, many studies have revealed that antiviral agents are effective against VZV-induced Ramsay–Hunt syndrome [138,139].

In one study using mice, the use of acyclovir after damage to the sciatic nerve and infection with HSV increased the thickness of nerve fibers and increased muscle re-innervation [140]. In another study, when acyclovir was administered to mice with HSV infection of the facial nerve, the incidence of facial paralysis decreased. The incidence of facial paralysis was lower in the high-dose group than in the low-dose group. In addition, the administration of acyclovir after facial paralysis exhibited faster improvement of facial paralysis, and high-dose administration was more effective [137] (Table 12).

## 3. Conclusions

While many drugs can aid in the recovery from facial nerve damage, none guarantee complete recovery. Therefore, several drugs that promote nerve regeneration have been studied. However, little is known about how these drugs restore the facial nerve. Although the functional examination of nerves presented in these studies can evaluate nerve regeneration, the morphological and histological findings may not be clinically correlated with nerve regeneration. If the physiological implications of these morphological and histological findings are determined in the future, it will be possible to better evaluate the effects of the drugs.

There is a limit to predicting the effects of nerve regeneration drugs administered to humans. Furthermore, it is not yet known when to begin administering drugs, what dosages to administer, and whether to administer a drug locally or systemically. In addition, the short- and long-term side effects and other systemic effects of these drugs require consideration. Therefore, it is difficult to use many of the drugs that have been presented in various clinical studies. Clinically, facial paralysis is a manifestation of various disease processes, and the treatment methods may vary depending on the underlying etiology.

The drugs summarized in this review are widely used in various clinical situations and are predictable, with relatively well-known administration methods and few or well-known side effects. However, further research is required to determine whether these drugs are effective in patients with facial nerve damage. Most of the studies in this review were conducted on nonhuman animals, in particular rodents, which have a different anatomy compared with that of humans. 

Since nerve regenerative ability may be greater in rodents than in humans after peripheral nerve damage, more studies as well as clinical trials are needed to determine whether the results of these studies can be applied to humans. A well-designed randomized controlled trial is essential to establish the use of a drug as a standardized treatment for facial nerve damage, such as steroid treatment, in patients with Bell’s palsy. In addition, research is also needed on the combination therapy of various drugs and on which drugs are appropriate according to the type, extent, and degree of facial nerve damage.

## Figures and Tables

**Table 1 biomedicines-10-01678-t001:** Comparison of glucocorticoids.

Glucocorticoid	Equivalent Dose (mg)	Relative Anti-Inflammatory Activity	Duration of Action (h)	Plasma Half-Life (h)
Short-acting				
Cortisone	25	0.8	8–12	0.5
Cortisol	20	1	8–12	1.5–2.0
Intermediate-acting				
Prednisone	5	4	12–36	3.4–3.8
Prednisolone	5	4	12–36	2.1–3.5
Methylprednisolone	4	5	12–36	>3.5
Triamcinolone	4	5	12–36	2–5
Long-acting				
Dexamethasone	0.75	20–30	36–72	3–4.5
Betamethasone	0.6	20–30	36–72	3–5

**Table 2 biomedicines-10-01678-t002:** Summary of studies on steroids for the treatment of peripheral nerve injury.

Drug	Author. Year [Reference]	Study Design	Species and/or Sample	Detection Method	Sample	Results/Conclusion
Dexamethasone	Lieberman et al. [15]	Animal study	C57BL/Ka Mice	Crush injury Intraperitoneal injection	Facial nerve	Low-dose dexamethasone (1 mg/kg/day) for 7 days enhanced functional recovery after injury, while a high dose (10 mg/kg/day) did not 28% decrease in total white blood cell count, 58% decrease in lymphocyte percentage, and 71% decrease in absolute lymphocyte count
Dexamethasone	Longur et al. [16]	Animal study	Wistar rats	Transection injury Intraperitoneal injection Group 1: controls Group 2: bumetanide Group 3: dexamethasone Group 4: bumetanide + dexamethasone	Facial nerve	Electroneurography latency difference in Group 1 was significantly higher than those in Groups 2–4. Electroneurography latency increases in Groups 2 and 3 were higher than that in Group 4 Higher axon number and intensity in Group 4 than in Groups 2 and 3
Dexamethasone	Jang et al. [17]	Animal study	Sprague–Dawley rats	Crush injury Topical dexamethasone	Facial nerve	Significantly lower recovery of the threshold of muscle action potentials in the experimental group than in the control group No statistical significance in nerve conduction velocity Dexamethasone treatment groups showed a larger axon diameter and thicker myelin sheath compared to the control group
Dexamethasone	Suslu et al. [18]	Animal study	Sprague–Dawley rats	Crush injury Intraperitoneal injection	Sciatic nerve	Statistically significant different changes in sciatic functional index measurements of all animals at days 7, 14, 21, and 28 The changes in the group treated with local dexamethasone were more remarkable than those in the group treated with systemic dexamethasone
Methylprednisolone	Ozturk et al. [19]	Animal study	Sprague–Dawley male rats	Crush injury Intraperitoneal injection Group I: ozone Group II: methylprednisolone Group III: ozone and methylprednisolone Group IV: isotonic saline	Sciatic nerve	Remarkably low degeneration in Group III, with no change in nerve sheath cells in Group II Degeneration, nerve sheath cell atrophy, intraneural inflammatory cellular infiltration, perineural granulation tissue formation, perineural vascular proliferation, perineural inflammatory cellular infiltration, and inflammation in peripheral tissue were observed
Methylprednisolone	Chao et al. [24]	Animal study	Wistar rats	Crush injury Topical dexamethasone	Facial nerve	Locally injected MP delivered by C/GP-hydrogel effectively accelerated facial functional recovery Regenerated facial nerves in the C/GP-MP group were more mature than those in the other groups The expression of GAP-43 protein was also improved by MP, particularly in the C/GP-MP group
Methylprednisolone	Mehrshad et al. [23]	Animal study	White Wistar rats	10-mm sciatic nerve defect was bridged using a chitosan conduit (CHIT/CGP-Hydrogel) filled with CGP-hydrogel or methylprednisolone (CHIT/MP)	Sciatic nerve	Faster recovery of regenerated axons in the methylprednisolone-treated group than in the CHIT/Hydrogel group
Methylprednisolone	Li et al. [25]	Animal study	Sprague–Dawley male rats	The anastomotic ends of the sciatic nerve were wrapped with a methylprednisolone sustained-release membrane. Comparison between methylprednisone alone or methylprednisone microspheres	Sciatic nerve	Methylprednisolone microsphere sustained-release membrane reduced tissue adhesion, inhibited scar tissue formation at the site of anastomosis, and increased the sciatic nerve function index and thickness of the myelin sheath
Methylprednisolone	Chen et al. [20]	Animal study	Guinea pig	Transection injury Intramuscular injection	Facial nerve	High-dose methylprednisolone elicited a delayed increase in nitric oxide formation and, thus, may concomitantly enhance the survival time of motor neurons after facial nerve transection
Methylprednisolone	Sevuk et al. [21]	Animal study	Female Wistar albino rats	Crush injury Intraperitoneal injection of high-dose methylprednisolone (30 mg/kg/day), and normal-dose methylprednisolone (1 mg/kg/day), and oral intake of vitamin A (10,000 IU/kg/day)	Sciatic nerve	Significantly lower serum nitric oxide and malondialdehyde levels after high-dose methylprednisolone, normal-dose methylprednisolone, high-dose methylprednisolone + vitamin A, normal-dose, and methylprednisolone + vitamin A treatment modalities compared to controls
Methylprednisolone	Yildirim et al. [22]	Animal study	New Zealand rabbits	Transection injury, compression injury, HSV type 1 infection Intramuscular injection	Facial nerve	In the group with a compressive lesion, axonal degeneration, myelin degeneration, and edema were significantly higher in the control group than in the methylprednisolone-treated group Among animals inoculated with Type 1 HSV, the treatment and control groups showed no significant differences in perineural fibrosis, axonal degeneration, myelin degeneration, or Schwann cell proliferation. The only statistically significant advantage of the treatment group was in edema formation
Prednisolone	Sullivan et al. [27]	Randomized, double-blind, placebo-controlled, factorial trial	Patients with Bell’s palsy	Patients recruited within 72 h after symptom onset Randomly assigned to receive 10 days of treatment with prednisolone, acyclovir, both agents, or placebo	Facial nerve	Early treatment with prednisolone significantly improved the chances of complete recovery at 3 and 9 months No evidence of a benefit of acyclovir alone or an additional benefit of acyclovir in combination with prednisolone
Prednisolone	Engström et al. [26]	Randomized, double-blind, placebo-controlled, multicenter trial	Patients with Bell’s palsy	Patients randomly assigned in permuted blocks of eight to receive placebo plus placebo; 60 mg prednisolone per day for 5 days then reduced by 10 mg per day plus placebo; 1000 mg valaciclovir three times per day for 7 days plus placebo; or prednisolone (10 days) plus valaciclovir (7 days)	Facial nerve	Significantly shorter time to recovery in the 416 patients who received prednisolone compared to the 413 patients who did not No difference in time to recovery between the 413 patients treated with valaciclovir and the 416 patients who did not receive valaciclovir
Dexamethasone	Galloway et al. [28]	Animal study	Sprague–Dawley rats	Crush injury Dexamethasone saturated gelfoam placed at the site of injury	Sciatic nerve	More rapid recovery in the steroid group at postoperative days 14, 18, and 22, which reached statistical significance at postoperative day 14
21-aminosteroid	Nasser et al. [29]	Animal study	Sprague–Dawley rats	Crush injury Intraperitoneal injection injections of 3 mg/kg U-74006F at 2-h intervals	Sciatic nerve	Significant improvement in motor function compared with the controls on days 14, 21, 25, and 28 for mature rats and on days 11 and 14 for immature rats
Betamethasone	Al-Bishri et al. [30]	Animal study	Wistar rats	Crush injury Subcutaneous injection betamethasone	Sciatic nerve	Short-term perioperative administration of betamethasone had a beneficial effect on the recovery of injured rat sciatic nerves

**Table 3 biomedicines-10-01678-t003:** Summary of studies assessing the use of statins for the treatment of peripheral nerve injury.

Drug	Author. Year [Reference]	Study Design	Species and/or Sample	Detection Method	Sample	Results/Conclusion
Simvastatin	Xavier et al. [37]	Animal study	Male Wistar rats	Crushing injury Intraperitoneal injection	Sciatic nerve	Simvastatin increased Sciatic Function Index scores and decreased areas of edema and mononuclear cell infiltration during Wallerian degeneration and nerve regeneration
Simvastatin	Guo et al. [38]	Animal study	Sprague–Dawley rats	Sciatic nerve defects in rats Chitosan conduit filled with 0, 0.5, or 1.0 mg simvastatin in Pluronic F-127 hydrogel	Sciatic nerve	Chitosan conduit filled with simvastatin/Pluronic F-127 hydrogel promoted nerve regeneration
Atorvastatin	Pan et al. [39]	Animal study	Sprague–Dawley rats	Crush injury Intake orally	Sciatic nerve	Atorvastatin improved damage-associated alterations, including structural disruption, oxidative stress, inflammation, and apoptosis
Atorvastatin	Cloutier et al. [40]	Animal study	Sprague–Dawley rats	Complete sciatic nerve section Intraperitoneal injection	Sciatic nerve	Better kinematics in atorvastatin-treated rats
Atorvastatin	Roselló-Busquets et al. [41]	In vitro and in vivo study	Microfluidic system and organotypic model	In vitro and in vivo in both the central and peripheral nervous systems	External granular layer cells as a central nervous system example, dorsal root ganglion neurons as a peripheral nervous system example	Cholesterol depletion promoted axonal growth in developing axons and increased axonal regeneration in vitro and in vivo both in the central and peripheral nervous systems

**Table 4 biomedicines-10-01678-t004:** Summary of studies assessing melatonin for the treatment of peripheral nerve injury.

Drug	Author. Year [Reference]	Study Design	Species and/or Sample	Detection Method	Sample	Results/Conclusion
Melatonin	Yanilmaz et al. [55]	Animal study	New Zealand rabbits	Transection injury Intraperitoneal injection	Facial nerve	In the nerve conduction study, the latent period was shortened but the amplitudes did not show a significant change in the melatonin group
Melatonin	Kaya et al. [56]	Animal study	Wistar rats	Transection injury, Crush injury Intraperitoneal injection	Sciatic nerve	Rats treated with melatonin showed better structural preservation of the myelin sheaths than the non-treated group Rats treated with melatonin also showed lower lipid peroxidation and higher superoxide dismutase, catalase, and glutathione peroxidase activities in sciatic nerve samples than the non-treated groups
Melatonin	Kaya et al. [57]	Animal study	Wistar rats	Crush injury Intraperitoneal injection	Sciatic nerve	Functional (sensory-motor, biochemical, and electrophysiological analyses) and morphological (light microscopic and ultrastructural analyses) data in the melatonin group showed beneficial effects of melatonin on axonal regeneration and functional recovery
Melatonin	Kaya et al. [58]	Animal study	Wistar rats	Transection injury Intraperitoneal injection	Sciatic nerve	Beneficial effect of melatonin in the light period. However, no significant beneficial effect of melatonin on recovery of the cut sciatic nerve in the dark period was observed The effect of melatonin on the recovery of the cut injured sciatic nerve depended on the time of treatment and may be attributed to the circadian rhythm
Melatonin	Guo et al. [59]	Animal study	Sprague–Dawley rats	C5–7 nerve roots were avulsed. The C6 nerve roots were then replanted to construct the brachial plexus nerve-root avulsion model Intraperitoneal injection	C5–7 nerve roots	Lower levels of C5–7 intramedullary peroxidase and malondialdehyde-melatonin combined with chondroitin sulfate ABC promoted nerve regeneration after nerve-root avulsion injury of the brachial plexus
Melatonin	Yazar et al. [60]	Animal study	Wistar albino rats	Compression injury Intraperitoneal injection	Sciatic nerve	A single injection of high-dose melatonin (100 mg/kg) preserved the myelin sheath, prevented axonal loss, and accelerated functional recovery during nerve regeneration in peripheral nerve injury
Melatonin	Stazi et al. [61]	Animal study	C57BL/6 mice	Transection injury, Compression injury Acute and reversible presynaptic degeneration induced by the spider neurotoxin α-Latrotoxin Intraperitoneal injection	Sciatic nerve	Melatonin promoted nerve terminal regeneration
Melatonin	Liu et al. [62]	Animal study	Male Wistar rats	End-to-side neurorrhaphy (ESN) Melatonin injection for 1 month	Musculocutaneous nerve	Melatonin treatment enhanced functional recovery after ESN compared to the recovery observed in the saline-treated group - Enhanced expression of GAP43 and b3-tubulin - Melatonin may promote functional recovery after peripheral nerve injury by accelerating cytoskeletal remodeling through the melatonin receptor-dependent pathway
Melatonin	Qian et al. [63]	Animal study	Sprague–Dawley rat Schwann cell (RSC)	Melatonin /polycaprolactone solution was sprayed onto a tubular mold cell counting kit 8 assay Immunofluorescent staining for actin, Ki67, S100, Tuj1, and MBP	Rat Schwann cell	Increased Schwann cell proliferation and neural expression in vitro and increased functional, electrophysiological, and morphological nerve regeneration in vivo

**Table 5 biomedicines-10-01678-t005:** Summary of studies that assessed the use of growth hormones for the treatment of peripheral nerve injury.

Drug	Author. Year [Reference]	Study Design	Species and/or Sample	Detection Method	Sample	Results/Conclusion
Growth hormone	Lopez et al. [71]	Animal study	Lewis rats	Transection injury Subcutaneous injection	Median nerve	Growth hormone-treated animals showed increased median nerve regeneration, as measured by axon density, axon diameter, and myelin thickness; improved muscle re-innervation; reduced muscle atrophy; and greater motor function recovery
Growth hormone	Saceda et al. [73]	Animal study	Wistar rats	Sectioning of the ulnar nerve in rats. The proximal and distal ends were sutured to either end of a silastic tube Subcutaneous injection	Ulnar nerve	The group receiving recombinant growth hormone showed improved recovery of conduction velocity, a more gradual increase in the amplitude of motor potential, improved architecture of the regenerating nerve, a greater nerve fiber density, and increased myelination with a lower degree of endoneural fibrosis
IGF-1	Nagata et al. [74]	Animal study	BALB/c albino mice	Cryo-injury IGF-1 was introduced into the muscle by hydrodynamic injection of IGF-1-expressing plasmid DNA using a biocompatible nonviral gene carrier, a polyplex nanomicelle	Sciatic nerve	IGF-1-expressing pDNA promoted early recovery of motor function IGF-1 also promoted early recovery of sensation after sciatic nerve injury
IGF-1	Peter et al. [75]	Animal study	Fischer 344 × Brown Norway rats	Transection injury The nerve stumps were placed at opposing ends of a custom-made T-tube, and the middle arm was attached to a minipump. An Alzet 2004 mini-osmotic pump (Durect Corp., Cupertino, California) delivered either normal saline or IGF-1 at a rate of 0.25 µL/h	Tibial nerve	IGF-1 increased the axon number, diameter, and density in regenerated nerves of both young and aged animals IGF-1 increased the myelination and Schwann cell activity in regenerated nerves of both young and aged animals IGF-1 preserved the morphology of postsynaptic neuromuscular junctions in aged animals

**Table 6 biomedicines-10-01678-t006:** Summary of studies assessing the use of carnitine for the treatment of peripheral nerve injury.

Drug	Author. Year [Reference]	Study Design	Species and/or Sample	Detection Method	Sample	Results/Conclusion
Acetyl-L-carnitine	Onger et al. [89]	Animal study	Wistar albino rats	Transection injury Intraperitoneal injection	Sciatic nerve	Carnitine had a beneficial effect on the regeneration of unmyelinated axons
Acetyl-L-carnitine	Hart et al. [90]	Animal study	Sprague- Dawley rats	Transection injury Intraperitoneal injection	Sciatic nerve	Neuroprotective effect of high-dose carnitine treatment was preserved after neuron loss
Acetyl-L-carnitine	Wilson et al. [91]	Animal study	Wistar rats	Transection injury Intraperitoneal injection	Sciatic nerve	Significantly higher mean number of myelinated axons in the carnitine group Greater mean myelin thickness in the carnitine group Carnitine also morphologically improved the quality of regeneration and target organ re-innervation
Acetyl-L-carnitine	Farahpour et al. [92]	Animal study	Wistar rats	Sciatic nerve defect was bridged using a chitosan conduit filled with 10 μL carnitine (100 ng/mL)	Sciatic nerve	Significant differences between muscle weight ratios. Significantly higher myelinated fiber number and diameter
Acetyl-L-carnitine	Mannelli et al. [93]	Animal study	Sprague–Dawley rats	Transection injury Cytochrome C (cytosolic fraction extraction) DNA fragmentation (Terminal deoxynucleotidyl transferase dUTP nick end labeling assay)	Sciatic nerve	Significantly decreased expression of the 19-kDa and 16-kDa fragments in a carnitine-treated group, which also showed significantly lower caspase 3 activity

**Table 7 biomedicines-10-01678-t007:** Summary of studies that assessed vitamin B12 for the treatment of peripheral nerve injury.

Drug	Author. Year [Reference]	Study Design	Species and/or Sample	Detection Method	Sample	Results/Conclusion
Vitamin B12	Tamaddonfard et al. [102]	Animal study	Wistar rats	Crush rush	Tibial nerve	Recovery of tibial function index values were significantly accelerated Wallerian degeneration was reduced,
Vitamin B12	Altun et al. [103]	Animal study	Wistar rats	Crush injury	Sciatic nerve	Tissue levels of vitamin B complex and vitamin B12 varied with progression of crush-induced peripheral nerve injury, and supplementation of these vitamins in the acute period may be beneficial for acceleration of nerve regeneration
Vitamin B12	Shibuya et al. [104]	Human study	Patients with immune-mediated or hereditary neuropathy	Intravenous injection	Sciatic nerve	Twelve patients were evaluated for the primary outcomes, which improved in seven patients and were unchanged or worsened in the remaining five
Vitamin B12	Gan et al. [105]	Animal study	ICR mice	Crush injury Intraperitoneal injection	Sciatic nerve	Vitamin B12 significantly improved functional recovery of the sciatic nerve, thickened the myelin sheath in myelinated nerve fibers, and increased the cross-sectional area of target muscle cells Furthermore, mecobalamin upregulated mRNA expression of growth-associated protein 43 in nerve tissue ipsilateral to the injury, and of neurotrophic factors (nerve growth factor, brain-derived nerve growth factor, and ciliary neurotrophic factor) in the L4–6 dorsal root ganglia
Vitamin B12	Okada et al. [106]	Animal study	Wistar rats	Transection injury Subcutaneous injection	Sciatic nerve	Vitamin B12 concentrations >100 nM promoted neurite outgrowth and neuronal survival; these effects were mediated by the methylation cycle, a metabolic pathway involving methylation reactions Vitamin B12 increased Erk1/2 and Akt activities through the methylation cycle In a rat sciatic nerve injury model, continuous administration of high doses of methylcobalamin improved nerve regeneration and functional recovery

**Table 8 biomedicines-10-01678-t008:** Summary of studies evaluating Ginkgo biloba for the treatment of peripheral nerve injury.

Drug	Author. Year [Reference]	Study Design	Species and/or Sample	Detection Method	Sample	Results/Conclusion
Ginkgo biloba	Zhu et al. [119]	Animal study	Sprague–Dawley rats	Cutting injury Intraperitoneal injection	Sciatic nerve	Ginkgo biloba significantly increased the number of myelinated fibers and the average diameter of the nerves within the graft
Ginkgo biloba	CH Jang et al. [122]	Animal study	Sprague–Dawley rats	Crush injury Intraperitoneal injection	Facial nerve	Ginkgo biloba significantly accelerated the recovery of vibrissae orientation
Ginkgo biloba	H Lin et al. [120]	Animal study	Sprague–Dawley rats	Transection injury Intake orally	Sciatic nerve	Sensory regeneration distance, sciatic functional index, motor nerve conduction velocity, compound muscle action potential, axon regeneration index, and muscle mass were significantly increased in the ginkgo biloba groups
Ginkgo biloba	Hsu et al. [121]	In vivo and in vitro study	Sprague–Dawley rats	Schwann cells in serum-deprived culture medium Different doses of ginkgo biloba (0, 1, 10, 20, 50, 100, 200 mg/mL)	Sciatic nerve Schwann cell	Thickened myelin sheath and increased cross-sectional area of target muscle cells Upregulated mRNA expression of growth-associated protein 43 in nerve tissue ipsilateral to the injury and neurotrophic factors in the L4-6 dorsal root ganglia

**Table 9 biomedicines-10-01678-t009:** Summary of studies assessing the use of coenzyme Q10 biloba for the treatment of peripheral nerve injury.

Drug	Author. Year [Reference]	Study Design	Species and/or Sample	Detection Method	Sample	Results/Conclusion
Coenzyme Q10	Yildirim et al. [125]	Animal study	Sprague–Dawley albino rats	Crush injury Intraperitoneal injection	Facial nerve	Significantly lower nerve stimulation thresholds in the coenzyme Q10 injection group Significant differences in vascular congestion, macrovacuolization, and myelin thickness between the coenzyme Q10 and control groups identified by light microscopy
Coenzyme Q10	Moradi et al. [126]	Animal study	Sprague–Dawley rats	Crush injury Intraperitoneal injection	Sciatic nerve	Faster recovery of regenerated axons in the coenzyme Q10 treatment group Regenerated fibers showed significantly higher myelinated fiber number and diameter in the coenzyme Q10 treatment group

**Table 10 biomedicines-10-01678-t010:** Summary of studies that used nimodipine to treat peripheral nerve injury.

Drug	Author. Year [Reference]	Study Design	Species and/or Sample	Detection Method	Sample	Results/Conclusion
Nimodipine	Zee et al. [127]	Animal study	Wistar rats	Crush injury Intake orally	Sciatic nerve	Oral administration of the Ca2+-entry blocker nimodipine accelerated the recovery of sensorimotor function in a dose-dependent manner
Nimodipine	Zee et al. [128]	Animal study	Wistar rats	Walking pattern analysis Oral intake	Walking pattern	Nimodipine delayed the onset of age-related motor deficits and could also counteract the deficits already present
Nimodipine	Angelov et al. [129]	Animal study	Wistar rats	Transection injury Food pellets containing 1000 ppm nimodipine	Facial nerve	Nimodipine accelerated axonal sprouting Nimodipine reduced the polyneuronal innervation of target muscles
Nimodipine	Scheller et al. [130]	Human study	Patients with a peripheral facial nerve paresis after maxillofacial surgery	House–Brackmann (HB) grade Intake orally	Facial nerve	Positive effect of nimodipine on the acceleration of peripheral facial nerve regeneration after surgically caused trauma
Nimodipine	Zheng et al. [131]	Animal study	Sprague–Dawley rats	Crush injury Oral intake	Facial nerve	Apparent recovery of electroconductivity. Higher amplitude and shorter latency time in the surgery plus nimodipine group compared to those in the surgery-only group Obvious recovery of myelination and reduction in the number of infiltrating cells in rats treated with nimodipine

**Table 11 biomedicines-10-01678-t011:** Summary of studies assessing ozone treatment of peripheral nerve injury.

Drug	Author. Year [Reference]	Study Design	Species and/or Sample	Detection Method	Sample	Results/Conclusion
Ozone	Ozbay et al. [133]	Animal study	Wistar albino rats	Crush injury Intraperitoneal injection	Facial nerve	Lower stimulation thresholds in the zone-treated group Significant differences in vascular congestion, macrovacuolization, and myelin thickness
Ozone	Ogut et al. [134]	Animal study	Wistar albino rats	Transection injury Intraperitoneal injection	Sciatic nerve	Significant differences in plasma superoxide dismutase, catalase, and glutathione peroxidase activities Significant functional improvement

**Table 12 biomedicines-10-01678-t012:** Summary of studies assessing antiviral treatment of peripheral nerve injury.

Drug	Author. Year [Reference]	Study Design	Species and/or Sample	Detection Method	Sample	Results/Conclusion
Acyclovir	Gumenyuk et al. [140]	Animal study	BALB/c line mice	Crush injury HSV-1 infection Intraperitoneal injection	Sciatic nerve	Acyclovir increased the nerve fiber thickness and muscle re-innervation
Acyclovir	Takahashi et al. [137]	Animal study	BALB/c AJcl mice	HSV-1 infection Intraperitoneal injection	Facial nerve	The incidence of facial nerve paralysis was significantly lower in the group given acyclovir before the paralysis than in the controls, and the duration of facial nerve paralysis was shorter

## Data Availability

Not applicable.

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
