# Peer review of "Review of Drug Therapy for Peripheral Facial Nerve Regeneration That Can Be Used in Actual Clinical Practice"

_biomedicines, 2022, doi:10.3390/biomedicines10071678_

Round 1
Reviewer 1 Report
This is a very well written comprehensive review of drugs that have shown efficacy for nerve regeneration and could be used off-label for facial nerve regeneration.
Intro:
"The next most important factor is the drug used." This seems presumptuous. Do you have a citation? Please remove or moderate this statement. The rest of the paragraph lays out the uncertainty of efficacy and limitations of use for the the various drugs that can be considered... other than steroids, none of these drugs have been clinically tested and therefore are not clinically indicated for facial nerve regeneration
2.3.2 Growth Hormones
-There are many more studies showing the efficacy of IGF-1 for nerve injuries in translational models. Consider including some of these.
Reviewer 2 Report
This subject is of great importance given the significant morbidity facial paralysis can cause and the little known regarding medical therapy in patients with facial paralysis. The authors do a good job presenting the broad and varied evidence existing for medical therapies in facial paralysis. I would recommend adding 1-2 more paragraphs to the conclusion to summarize the research presented and offer suggestions for future research. Specifically, the below topics should be presented in the conclusion:
1) Majority of facial nerve medical therapy research has been performed in rodents. Rodents have robust peripheral nerve regeneration, likely at least in part due to the short distances axons have to regenerate across compared to humans. More studies are required in NHP models, as well as more clinical trials.
2) Much of this research does not assess functional outcomes but presents morphologic and histologic findings that may or may not correlate with improved function.
3) Clinically, facial paralysis is a manifestation of many different disease processes. Medications may have differing impacts on facial nerve recovery depending on the underlying etiology.
Another moderate revision required is a section on Antiviral therapy as Valacyclovir and Acyclovir are relatively well-studied medications in Ramsay Hunt syndrome as well as Bell's palsy. This is the second most commonly used medication clinically in facial paralysis patients and so should be discussed.
Minor revisions suggested:
1) The abstract mentions complete recovery after facial paralysis is "impossible." This statement is not correct as many patients after Bell's palsy, Ramsay Hunt syndrome, and other causes of facial paralysis make a complete recovery.
2) Page two notes the second most important factor regarding how well a patient will recover from facial paralysis is the "drug used." This statement is not correct (depends on the etiology of facial paralysis, duration of facial paralysis, etc.) and so I would recommend removing it or adjusting to say an important factor regarding outcome of paralysis is drug used.
